# Depth of Bacterial Penetration into Dentinal Tubules after Use of Different Irrigation Solutions: A Systematic Review of In Vitro Studies

Igor Tsesis [†] , Michal Lokshin *,[†] , Dan Littner , Tomer Goldberger [‡] and Eyal Rosen [‡]

Department of Endodontology, Maurice and Gabriela Goldschleger School of Dental Medicine,
Sackler Faculty of Medicine, Tel Aviv University, Tel Aviv 6997801, Israel
* Correspondence: michal.loksh@gmail.com
† These authors contributed equally to this work.
‡ These authors contributed equally to this work.

**Abstract:** (1) The objective of the study is to conduct a comprehensive systematic review of in vitro studies in order to assess the depth to which *E. faecalis* bacteria penetrate human dentinal tubules after the use of various irrigation solutions. (2) Methods: A literature search of the MEDLINE, Scopus, Cochrane CENTRAL, and Embase databases was conducted, as well as a backward and forward citation search. Two independent reviewers then selected suitable studies based on inclusion and exclusion criteria. Data were extracted and the risk of bias and methodology of the studies were evaluated. (3) Results: Out of a total of 504 papers evaluated following the removal of duplicates, 7 studies met the inclusion criteria and were included in the systematic review. The heterogeneity of the studies made it impossible to perform a meta-analysis. The majority of the studies reported that sodium hypochlorite (NaOCl) and chlorhexidine digluconate (CHX) can affect the penetration depth of *E. faecalis* suspensions. The studies included in this review possess a moderate to high risk of bias and thus represent moderate evidence that the antimicrobial activity of NaOCl and CHX affects the intra-tubular penetration of bacteria. (4) Conclusions: The evidence indicates that irrigants may affect the bacteria inside human dentinal tubules. Standardized high-quality methods are needed to evaluate bacterial penetration in in vitro studies.

**Keywords:** endodontic infection; bacterial penetration; endodontic irrigants; dentinal tubuli

## 1. Introduction

Apical periodontitis (AP) involves inflammation and destruction of peri-radicular tissues. The condition may be the result of pulp infection or physical or iatrogenic trauma to the pulp and extrusion of root canal filling materials [1]. Peri apical inflammation stimulates bone resorption and the formation of granulomas and cysts. In addition, immune responses are elicited in response to bacteria and their products [2] since there is a well-established relationship between periapical inflammation and bacterial infection [3–5]. The root canal wall is a surface to which bacteria can adhere, and a biofilm can develop when the root canal system is infected [6,7]. Such a biofilm provides microorganisms with a series of advantages and skills not available to individual cells living in a free-floating (planktonic) state [8] because the microorganisms are protected from chemical and mechanical stresses by the extracellular polymeric substance (EPS) [9,10].

Endodontic therapy is designed to cure or prevent apical periodontitis [11], with the goal of preserving the tooth as a functional unit within a functioning dentition. Bio-chemical preparation of the canal, the use of irrigation solutions, and a void-free, three-dimensional, hermetic root canal filling that minimizes the opportunity for reinfection [12] are three factors used in the repair of apical periodontitis [13].

Unfortunately, while free-living planktonic bacteria inside the root canal cavity are eradicated early in the process of cleaning and shaping the canal, bacteria in biofilms or less accessible areas are difficult to eradicate and may cause persistent disease [14]. If the root canal is infected, a biofilm may also be present in the lateral canals, fins, isthmuses, and dentinal tubules [15].

The goal of evidence-based dentistry (EBD) is to integrate the findings of high-quality clinical research into clinical decision making in order to optimize patient outcomes [16,17]. This study utilized evidence-based dentistry principles to conduct a systematic search and analysis of the available literature regarding bacterial penetration into dentinal tubules. In daily clinical practice, the elimination of bacteria within the root canal system is the most challenging aspect of root canal treatment. Endodontic irrigants play an important role in the elimination of bacteria. Various irrigation protocols have been purposed for this purpose without a necessary standardization. This review identifies the need to develop standardized methods to evaluate bacterial penetration depth after the use of different irrigants.

The aim of this study was to investigate the depth to which bacteria penetrate human dentinal tubules in vitro, following the use of various irrigation solutions.

## 2. Materials and Methods

### 2.1. Criteria for Considering Studies for This Review

This systematic review was conducted according to the PROSPERO guidelines [18]. Only studies that met the inclusion criteria were evaluated.

Inclusion criteria:

1. Full text in English.
2. In vitro study to evaluate how sodium hypochlorite (NaOCl) and chlorhexidine digluconate (CHX) affect bacterial penetration into human dentinal tubules.
3. The studies involve human intact, closed apex, single-canal rooted teeth. For this purpose, intact is defined as caries-free, with no resorption, and without fracture lines, anatomic irregularities, previous endodontic treatment, or curvatures.
4. Studies include a control group using saline irrigation or no irrigation.
5. Single strain *E. faecalis* (ATCC 29212) bacteria.
6. One irrigation material per study group (NaOCl or CHX) after smear layer removal with EDTA.
7. Use of one of the following methods to compare bacterial penetration/survival in the dentinal tubules: scanning electron microscopy (SEM), dentinal shavings, or confocal laser scanning microscopy (CLSM).

Exclusion criteria:

1. Full text not in English.
2. In vivo studies, case reports, and reviews.
3. Not human mature, intact teeth or multirooted, open apex, human teeth.
4. No control group.
5. Multispecies bacteria or not *E. faecalis* (ATCC 29212).
6. Multiple irrigation materials used in a study group.
7. Evaluation of colony-forming units (CFU) with paper points.

### 2.2. Search Methods for Identification of Studies

The literature search was performed following PRISMA guidelines [19].

The databases were searched up to October 2021. MEDLINE was searched with the PubMed search engine, www.ncbi.nlm.nih.gov (accessed on 5 October 2021). The other databases were Scopus, (www.scopus.com, accessed on 5 October 2021), Embase (www.embase.com, accessed on 5 October 2021), and Cochrane Central (www.cochranelibrary.com, accessed on 5 October 2021).

The population, intervention, comparison, and outcome (PICO) strategy was

1. Population: Extracted mature permanent human teeth undergoing root canal preparation using needle irrigation.
2. Intervention: NaOCl or CHX irrigation solutions used after the removal of the smear layer by EDTA.
3. Comparisons: Extracted mature permanent human teeth with complete root formation undergoing root canal preparation either without irrigation or with irrigation with saline.
4. Outcome: Depth of bacterial penetration into the dentinal tubules.

The search terms were: dentin OR dentinal tubule OR dentinal tubule penetration OR root canal AND biofilm OR *Enterococcus faecalis* OR *E. faecalis* OR bacterial penetration OR bacteria penetration AND irrigant OR needle irrigation OR saline OR NaOCl OR sodium hypochlorite OR saline AND confocal OR SEM OR scanning electron microscopy OR dentinal shavings OR CFU.

The search was limited to the English language. The bibliographies of included articles were searched manually in order to identify extra studies not detected by the electronic search.

MeSH received the following: (("dentin"[MeSH Terms] OR "dentin"[All Fields] OR "dentine"[All Fields] OR "dentines"[All Fields] OR "dentins"[All Fields] OR "dentin s"[All Fields] OR "dentinal"[All Fields] OR "dentine s"[All Fields] OR (("dentin"[MeSH Terms] OR "dentin"[All Fields] OR "dentine"[All Fields] OR "dentines"[All Fields] OR "dentins"[All Fields] OR "dentin s"[All Fields] OR "dentinal"[All Fields] OR "dentine s"[All Fields]) AND ("tubule"[All Fields] OR "tubule s"[All Fields] OR "tubular"[All Fields] OR "tubules"[All Fields])) OR (("dentin"[MeSH Terms] OR "dentin"[All Fields] OR "dentine"[All Fields] OR "dentines"[All Fields] OR "dentins"[All Fields] OR "dentin s"[All Fields] OR "dentinal"[All Fields] OR "dentine s"[All Fields]) AND ("tubule"[All Fields] OR "tubule s"[All Fields] OR "tubular"[All Fields] OR "tubules"[All Fields]) AND ("penetrability"[All Fields] OR "penetrable"[All Fields] OR "penetrate"[All Fields] OR "penetrated"[All Fields] OR "penetrates"[All Fields] OR "penetrating"[All Fields] OR "penetration"[All Fields] OR "penetrations"[All Fields])) OR "root canal"[All Fields]) AND ("biofilm s"[All Fields] OR "biofilmed"[All Fields] OR "biofilms"[MeSH Terms] OR "biofilms"[All Fields] OR "biofilm"[All Fields] OR *"Enterococcus faecalis"*[All Fields] OR *"E. faecalis"*[All Fields] OR "bacterial penetration"[All Fields] OR "bacteria penetration"[All Fields]) AND ("irrigant"[All Fields] OR "irrigants"[All Fields] OR "irrigate"[All Fields] OR "irrigated"[All Fields] OR "irrigates"[All Fields] OR "irrigating"[All Fields] OR "irrigational"[All Fields] OR "irrigator"[All Fields] OR "irrigators"[All Fields] OR "therapeutic irrigation"[MeSH Terms] OR ("therapeutic"[All Fields] AND "irrigation"[All Fields]) OR "therapeutic irrigation"[All Fields] OR "irrigation"[All Fields] OR "irrigations"[All Fields] OR "needle irrigation"[All Fields] OR ("saline solution"[MeSH Terms] OR ("saline"[All Fields] AND "solution"[All Fields]) OR "saline solution"[All Fields] OR "saline"[All Fields] OR "salines"[All Fields]) OR "NaOCl"[All Fields] OR ("sodium hypochlorite"[MeSH Terms] OR ("sodium"[All Fields] AND "hypochlorite"[All Fields]) OR "sodium hypochlorite"[All Fields]) OR ("saline solution"[MeSH Terms] OR ("saline"[All Fields] AND "solution"[All Fields]) OR "saline solution"[All Fields] OR "saline"[All Fields] OR "salines"[All Fields])) AND ("confocal"[All Fields] OR "confocally"[All Fields] OR ("struct equ modeling"[Journal] OR "scan electron microsc"[Journal] OR "sem"[All Fields]) OR "scanning electron microscopy"[All Fields] OR "dentinal shavings"[All Fields] OR "CFU"[All Fields])) AND (English[Filter]).

*2.3. Data Collection and Analysis*

2.3.1. Selection of Studies

Two reviewers (I.T. and M.L.) independently scanned the titles and abstracts to identify relevant studies, which were then subjected to full-text evaluation. In case of disagreement, the study was discussed until a consensus between the two reviewers was reached. We

extracted and analyzed the data and evaluated the methodological quality of articles deemed suitable for the study.

### 2.3.2. Data Extraction

Data were extracted by two observers (I.T and M.L) independently, and the following methodological parameters were recorded: authors and date of publication; the amount of *E. faecalis* incubated with each sample; bacterial strain and incubation time; solution in which teeth were stored until experiment; sample size; external surface treatment; smear layer removal solutions; irrigation protocol for the study group; the amount of irrigant used for study groups; the presence of a control group.

### 2.3.3. Risk of Bias and Quality Evaluation

The selected studies were assessed for risk of bias with parameters adapted from Sarkis-Onofre [20], Montagner [21], and AlShwaimi [22] to suit this review. The parameters included were: randomization of teeth, presence of control, description of sample size calculation, materials used according to the manufacturer's instructions, samples prepared by a single operator, use of single canal teeth free of caries, restoration, and resorptions, and blinding of the observer. A Y (yes) was scored if the parameter was reported in the article. If a parameter could not be found in the article, it received an N (no). The risk of bias was classified according to the numbers of Yes with a score of high (1–3 Yes), moderate (4–5 Yes), or low (6–7 Yes). Two reviewers assessed the articles independently and in the event of a dispute, the reviewers re-evaluated the article together.

### 2.3.4. Statistical Analysis

Our statistical approach was to assess the heterogeneity between studies by considering the I2 value, which indicates low, medium, or high heterogeneity. Similarities between the included studies were used to perform a meta-analysis.

## 3. Results

### 3.1. Literature Search

The databases search resulted in 963 articles. Removing duplicates left a total of 504 studies, of which 453 studies that were deemed not relevant to the topic of the current study were excluded. The 51 remaining full-text articles were evaluated to provide 7 articles for inclusion in the systematic review [23–29] (Figure 1).

### 3.2. Study Characteristics

The sample size in the seven studies analyzed ranged from 10 to 30 teeth per study group. None of the studies justified the sample size selection. Moreover, only one study (14%) reported using materials according to the manufacturer's instructions [25]. None of the studies blinded the observer and only in two studies (29%) were the samples prepared by a single operator [24,26].

There was no standardization between the studies regarding the number of bacteria inoculated into the canals or the incubation time. In three studies [23,24,28], bacteria were incubated for 21 days, whereas in three other studies, bacteria were incubated inside the dentinal tubules for 4 weeks [25–27] or 14 days [29]. In addition, there were differences in the amount of irrigant used and the size of the irrigation needle.

Authors in some studies evaluated the quantitative penetration of bacteria by CLSM [24,25], while the majority used colony-forming units (CFU) per milliliter in a known diameter of dentinal shavings [23,27–29]. One article used both methods of evaluation [26]. For two studies [24,28], when no answer to e-mail queries was received, we were forced to extract the data from graphs presented in the publication.

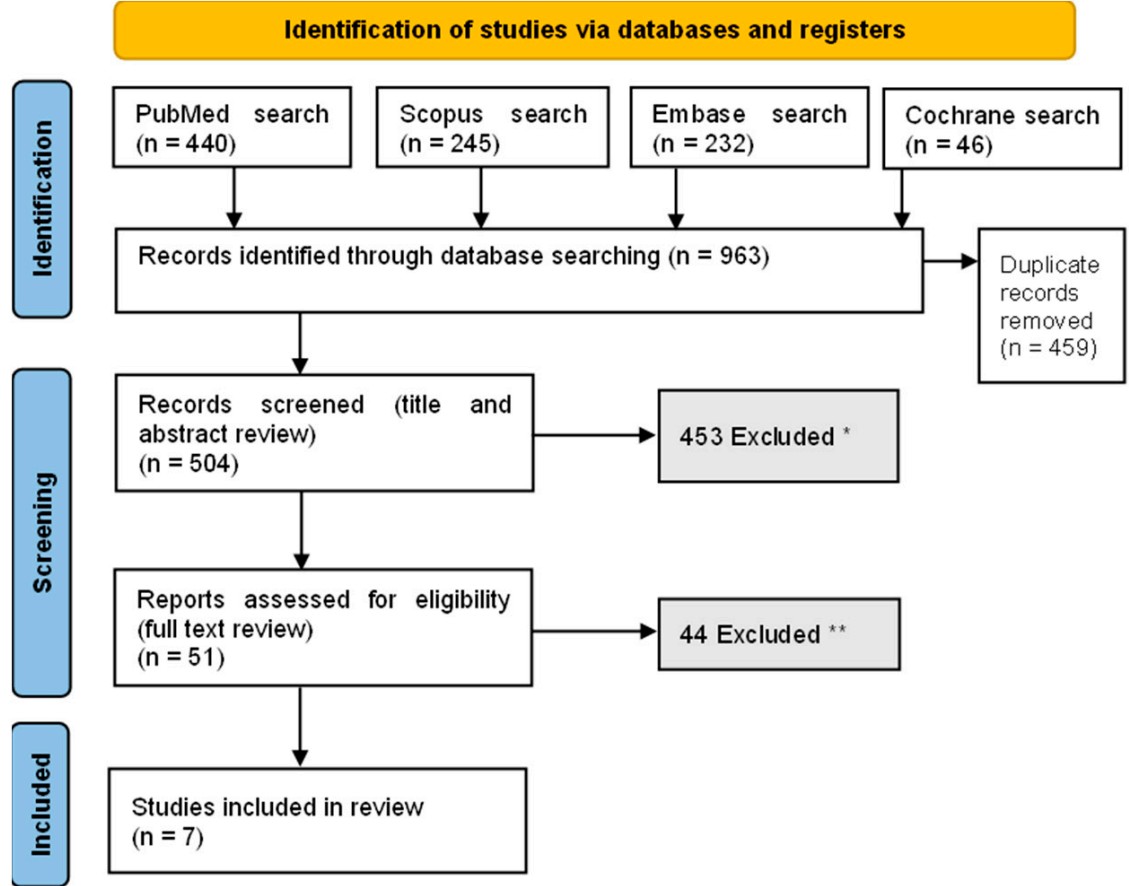

**Figure 1.** Flow chart of the literature search and the study selection process.

Endodontic treatment procedures reported in the studies varied with respect to the type of instrumentation technique, instruments that were used for preparation, and the concentration of sodium hypochlorite used as an irrigant. Teeth were stored in different solutions for different amounts of time before use, and different methods for smear layer removal were employed. The majority of the included studies (four studies, 57%) applied nail varnish to the outer surface of the prepared teeth in order to prevent bacterial leakage [23,24,28,29]. The characteristics of all included studies are presented in Table 1.

*3.3. Risk of Bias*

All included studies were assessed for risk of bias. Of the seven studies analyzed, only two studies (29%) qualify as having a moderate risk of bias [25,26], while the others (five studies, 71%) have a high risk of bias [23,24,27–29]. The results are described in Table 2 according to the parameters considered in the analysis.

### 3.4. Statistical Results

Because of the heterogeneity of the studies, it was not possible to perform a meta-analysis of the results. However, we did conduct a best-evidence synthesis consisting of five levels of evidence, as follows [30]:

1.  Strong evidence: Provided by two or more studies with high quality and/or generally consistent findings in all studies (>75% of the studies reported consistent findings);
2.  Moderate evidence: Provided by one study with high quality and/or two or more studies with low quality and generally consistent findings in all studies (75% of the studies reported consistent findings);
3.  Limited evidence: Provided by only one study with low quality;
4.  Conflicting evidence: Inconsistent findings in multiple studies (<75% of the studies reported consistent findings);
5.  No evidence: No relevant studies could be found.

Descriptive statistics were used to describe and summarize the reported outcomes of the studies. The results of bacterial penetration were summarized according to the quantitative method that was used.

### 3.5. Depth of Bacterial Penetration

The majority of the studies reported that NaOCl and CHX can affect the depth to which *E. faecalis* suspensions penetrate. The control group was saline irrigation or no irrigation.

#### 3.5.1. Confocal Laser Scanning Microscopy (CLSM)

Three studies (43%) used CLSM to evaluate bacterial penetration. Two studies (29%) assessed the percentage of live/dead bacteria inside dentinal tubules [24,26], while one study (14%) compared the bacterial dead zone (ZDB) in microns [25] with 5.25% NaOCl and 2% CHX used as irrigants, to control values with 0.9% normal saline. The results indicated significantly higher numbers of bacteria inside the dentinal tubes when normal saline was used compared to other irrigants. There were no significant differences in the ZDB between NaOCl and CHX. Although there were differences in the methods and measured parameters, specifically in the time, tooth area, and concentration of the irrigant, there was moderate evidence that the antimicrobial activity of NaOCl and CHX can decrease the intra-tubular penetration of bacteria. Table 3 summarizes the depth of penetration assessed by CLSM in the analyzed studies.

#### 3.5.2. Dentinal Shavings

Five studies (71%) evaluated dentin samples of varied thickness, taken from the inner surface of the root canal with different sizes of burs [23,26–29]. The CFU/milliliter was assessed in an area of 200–1000 microns. Two studies (29%) evaluated CFU over time, where one study [23] made the assessments after 1, 5, and 10 min, and the second [29] evaluated the CFU/milliliter at 0, 7, 14, 21, and 28 days. The results provide moderate evidence that the antimicrobial activity of sodium hypochlorite and chlorhexidine can reduce the intra-tubular penetration of bacteria as measured by this method of quantifying the bacteria. Table 4 summarizes the depth of penetration using dentinal shavings in the included studies.

#### 3.5.3. Scanning Electron Microscopy

Two studies [23,26] observed the tooth specimens by SEM to verify contamination with *E. faecalis* bacteria. This method of evaluation is qualitative and can only detect the presence or absence of bacteria/smear layer. As a result, this method was not included in our review as a quantitative method able to evaluate the percentage of live/dead bacteria in the dentinal tubules.

## 4. Discussion

This study was designed to review the depth to which bacteria penetrate into human dentinal tubules after needle irrigation with NaOCl or CHX. *E. faecalis*, the bacteria assessed, is strongly associated with persistent infections and endodontic treatment failure [31,32]. The root canal wall represents a surface to which bacteria can adhere and where a biofilm can thrive [6]. Bacteria tend to organize in biofilms and studies have shown the presence of varied microbiota in endodontic infections [33–36]. Today, alongside chemical irrigation, mechanical shaping is still one of the main ways of root canal disinfection, as it allows the use of an easy and quick three-dimensional obturation technique [37].

Only studies that evaluated human teeth were included in the present review [38]. A study with an in vitro model of dentinal tubule infection of bovine incisors with *E. faecalis* introduced in 1987 [38] reported that bacteria could invade the tubules from the pulpal and outer side of the tooth. The basic morphology of bovine teeth is similar to that of human teeth [39], although Hals and Olsen reported in 1984 [40] that bovine teeth have giant tubules, which are likely to affect the depth of bacterial penetration and the time it takes for bacteria to reach full dentin thickness. In this context, bacteria were found to penetrate 300–400 μm inside the canal after only one day of incubation. After 3 weeks of incubation, there was a dense infection at a depth of 300–400 μm and a moderate infection at 400–500 μm, while the front of the infection could reach 800–1000 μm.

CLSM and analysis of dentinal shavings analysis are two methods that can be used to obtain a quantitative assessment of the penetration depth of bacteria inside dentinal tubules. When the specimens are stained with a LIVE/DEAD BacLight bacterial viability kit, CLSM imaging can differentiate between live green bacteria and dead red bacteria. This method can count the bacteria along the dentinal tubules [41] and provide the percentage of dead bacteria at a specific depth, as well as provide information about biofilm structure and organization inside the dentinal tubules following treatment.

Determination of colony-forming units is a method with high variability because the use of a logarithmic scale means this method can only estimate the number of bacteria at depths determined by the bur diameter, but cannot provide exact cell numbers [41,42].

There are a number of studies [43–46] in which antibacterial assessments were made by flushing the canal with saline and inserting sterile paper points to absorb the contents inside the canal. The CFUs that can be grown on these paper points are then counted. We did not include such studies in our review because this method evaluates the CFU of planktonic bacteria inside the root canal space, and planktonic bacteria do not represent the clinical or in vivo situation where bacteria tend to invade the tubules and to be organized in a biofilm.

SEM is a good qualitative way to assess the presence or absence of bacteria and smear layer, but it does not provide a quantitative analysis [23,24]. Thus, studies that used SEM as the sole method of evaluation also were not included in the present study.

The studies that were included in this review are highly heterogeneous in design, with no standardization of the methods used. Each study incubated the bacteria using a different method and for a different period, used variable amounts of irrigants, irrigated the specimens with different size irrigation needles, and stored the teeth in different solutions, which can affect the penetrability and characteristics of the dentin specimens. All these varying parameters made it very difficult to compare the findings, and it was not feasible to conduct a meta-analysis. We suggest creating a standard for future studies designed to assess in vitro bacterial penetration.

We conducted a comprehensive quality assessment of the selected papers by adopting relevant parameters from previous reviews and meta-analyses that assessed in vitro studies [20–22]. Different protocols have been purposed for risk of bias evaluation in in vitro studies. We used a combination of three methods to standardize the bias assessment in order to provide the most comprehensive way to assess the risk of bias for this article. The analysis indicated that none of the studies blinded the observer nor calculated the sample size, which increases the risk of bias. In consequence, the results should be interpreted

with caution. Although randomized controlled clinical trials provide the most reliable results, well-designed in vitro studies with high-quality methodological and a low risk of bias could also provide valuable information for clinical situations [20,21,47].

This review identifies the need to develop standardized methods to evaluate bacterial penetration depth after the use of different irrigants. It is important to note that we only assessed studies that used needle irrigation, with no sonic or ultrasonic agitation. While it could introduce a certain degree of bias, similar protocols of irrigation should be compared in order to prevent the heterogeneity of the evaluated studies in the systematic review [48]. In the future, it would be of interest to examine how the method of irrigation affects the penetration depth of endodontic pathogens and sealers into the tubules.

## 5. Conclusions

Within the limitations of this review, the evidence indicates that the choice of irrigant can affect the bacteria inside human dentinal tubules and influence the depth to which bacteria penetrate human dentinal tubules as compared to saline irrigation or no irrigation. Standardized high-quality methods to evaluate bacterial penetration in in vitro studies are needed.

**Author Contributions:** Conceptualization, I.T., T.G. and D.L.; methodology, I.T.; validation, I.T.; formal analysis, I.T., T.G. and E.R.; investigation, I.T., M.L. and D.L.; resources, I.T.; data curation, M.L.; writing—original draft preparation, M.L.; writing—review and editing, I.T., M.L. and E.R.; visualization, M.L.; supervision, I.T. and E.R.; project administration, I.T. All authors have read and agreed to the published version of the manuscript.

**Funding:** This research received no external funding.

**Institutional Review Board Statement:** Not applicable.

**Informed Consent Statement:** Not applicable.

**Data Availability Statement:** Not applicable.

**Conflicts of Interest:** The authors declare no conflict of interest.

## Appendix A

**Table 1.** Characteristics of included studies.

| Study | Amount of *E. faecalis* | Type of Bacteria + Incubation Time | Teeth Stored in a Solution Until Experiment? | Total Number of Examined Teeth (Sample Size) | External Surface Treatment | Smear Layer Removal Solutions | Irrigation Study Group | Amount of Irrigant in Study Groups | Control Group |
|---|---|---|---|---|---|---|---|---|---|
| Parolia et al., 2021 [23] | 0.5 McFarland standard ($1.5 \times 10^8$ CFU/mL) | *E. faecalis* (ATCC 29212), incubated for 21 days | Stored in saline | 90 | Covered with nail varnish | Sonic irrigation with 5.25% NaOCl and then 17% EDTA for 2 min + 5% sodium thiosulfate for 10 min | (a) 2% CHX (n = 30); (b) 6% NaOCl (n = 30) | 5 mL using a 30-gauge side vented needle | Saline (n = 30) |
| Zeng et al., 2018 [24] | $1 \times 10^8$ cells/mL. 20 µL of bacterial suspension was inoculated into each canal | *E. faecalis* (ATCC 29212), incubated for 21 days | 0.9% (*w*/*v*) NaCl containing 0.02% sodium azide at 4 °C | 21 | Covered with two layers of nail varnish | 3% NaOCl + 2 mL of 17% EDTA | 3% NaOCl (n = 15) | 1.5 mL via a 30-gauge needle tip | No irrigation (n = 6) |
| Vatkar et al., 2016 [25] | 0.5 McFarland standard. 3 mL BHI liquid medium was added to each of the test tubes | *E. faecalis* (ATCC 29212), incubated for 4 weeks at 37 °C | Stored in saline | 40 | Not mentioned | Ultrasonically activated with aqueous EDTA for 4 min, then washed with sterile water | (a) 5.25% NaOCl (n = 10); (b) 2% CHX (n = 10) | Amount not mentioned, using a 23-gauge hypodermic needle and syringe | (a) No irrigation (n = 10); (b) 0.9% saline (n = 10) |
| Neelakantan et al., 2015 [26] | 3 mL *E. faecalis* suspension ($1 \times 10^8$ mL$^{-1}$) | *E. faecalis* (ATCC 29212), incubated under anaerobic conditions for 4 weeks at 37 °C | 0.01% sodium hypochlorite solution | 50 | Not mentioned | Ultrasonic bath of 5.25% sodium hypochlorite and 17% EDTA for 4 min each, rinsed in sterile water for 1 min | 3% NaOCl (n = 25) | Not mentioned | Saline (n = 25) |
| Ashofteh et al., 2014 [27] | 1 McFarland standard ($3 \times 10^8$ CFU/mL) | *E. faecalis* (ATCC 29212), incubated for 4 weeks at 37 °C | Not mentioned | 70 | Not mentioned | 1.3% NaOCl and 17% EDTA, added to the canals for 1 min and then all canals were rinsed with 5 mL of saline | (a) 5.25% NaOCl (n = 30); (b) 2% CHX (n = 30) | Amount not mentioned, using a 28-gauge needle | Saline (n = 10) |
| Nourzadeh et al., 2017 [28] | 0.5 McFarland standard ($1.5 \times 10^8$ CFU/mL). 2 mL of the bacterial inoculum | *E. faecalis* (ATCC 29212), incubated for 21 days at 37 °C | Stored in saline | 75 | Covered with nail polish | Ultrasonic bath with 17% EDTA for 10 min, followed by 5.25% NaOCl for 10 min and tap water for 1 h | (a) 5.25% NaOCl (n = 15); (b) 2.5% NaOCl (n = 15); (c) 2% CHX (n = 15); (d) 0.2% CHX (n = 15) | 2 mL of each irrigant using a 29 gauge needle, and then 4 mL of saline | Saline (n = 15) |
| Mohammadi et al., 2008 [29] | Not mentioned | *E.faecalis* (ATCC 29212), incubated for 14 days at 37 °C | 0.5% sodium | 70 | Covered with two layers of nail varnish | 5.25% NaOCl and 17% EDTA (with pH 7.2) | (a) 2% CHX (n = 30); (b) 2.6% NaOCl (n = 30) | 5 mL with sterile 3-mL plastic syringes and 27-gauge needles until dentin tubes were totally filled | No irrigation (n = 10) |

**Table 2.** Risk of bias of the included studies.

| Study | Randomization | Control | Sample Size Calculation | Materials Used According to Manufacturers' Instructions | Single Operator | Single Canal Teeth Free of Caries, Restoration and Resorptions | Blinding of the Observer | Risk of Bias |
|---|---|---|---|---|---|---|---|---|
| Parolia et al., 2021 [23] | Y | Y | N | N | N | Y | N | High |
| Zeng et al., 2018 [24] | N | Y | N | N | Y | Y | N | High |
| Vatkar et al., 2016 [25] | Y | Y | N | Y | N | Y | N | Moderate |
| Neelakantan et al., 2015 [26] | Y | Y | N | N | Y | Y | N | Moderate |
| Ashofteh et al., 2014 [27] | N | Y | N | N | N | Y | N | High |
| Nourzadeh et al., 2017 [28] | Y | Y | N | N | N | Y | N | High |
| Mohammadi et al., 2008 [29] | Y | Y | N | N | N | Y | N | High |

**Table 3.** Summary of CLSM results.

| Study | Study Group | Control Group | 50 µm | 100 µm | 150 µm | 200 µm | 300 µm | 400 µm | 500 µm | 600 µm | 1000 µm |
|---|---|---|---|---|---|---|---|---|---|---|---|
| Zeng et al., 2018 [24] | 3% NaOCl (n = 15) | No irrigation (n = 6) | Y | Y | Y | N | N | N | N | N | N |
| Vatkar et al., 2016 [25] | (a) 5.25% NaOCl (n = 10); (b) 2% CHX (n = 10) | (a) No irrigation (n = 10); (b) 0.9% Saline (n = 10) | N | Y | Y | N | N | N | N | N | N |
| Neelakantan et al., 2015 [26] | 3% NaOCl (n = 25) | Saline (n = 25) | N | N | N | Y | N | Y | N | N | N |

**Table 4.** Summary of dentinal shavings results.

| Study | Study Group | Control Group | 50 µm | 100 µm | 150 µm | 200 µm | 300 µm | 400 µm | 500 µm | 600 µm | 1000 µm |
|---|---|---|---|---|---|---|---|---|---|---|---|
| Parolia et al., 2021 [23] | (a) 6% NaOCl (n = 30); (b) 2% CHX (n = 30) | Saline (n = 30) | N | N | N | Y | N | Y | N | N | N |
| Neelakantan et al., 2015 [26] | 3% NaOCl (n = 25) | Saline (n = 25) | N | N | N | Y | N | Y | N | N | N |
| Ashofteh et al., 2014 [27] | (a) 5.25% NaOCl (n = 30); (b) 2% CHX (n = 30) | Saline (n = 10) | N | N | N | N | N | N | N | N | Y |
| Nourzadeh et al., 2017 [28] | (a) 2.5% NaOCl (n = 15); (b) 5.25% NaOCl (n = 15); (c) 0.2% CHX (n = 15); (d) 2% CHX (n = 15) | Saline (n = 15) | N | N | N | Y | N | Y | N | Y | N |
| Mohammadi et al., 2008 [29] | (a) 2.6% NaOCl (n = 30); (b) 2% CHX (n = 30) | No irrigation (n = 10) | N | N | N | N | N | N | Y | N | N |

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
