# Peer review of "Depth of Bacterial Penetration into Dentinal Tubules after Use of Different Irrigation Solutions: A Systematic Review of In Vitro Studies"

_applsci, doi:10.3390/app13010496_

Round 1

Reviewer 1 Report

Dear Authors, 

you made a great work!

However, some improvements are mandatory before acceptance. 

Author Response

- I believe that the introduction can be enriched from some points of view. In particular, the complete seal of the endodontic system, and not just a correct two-dimensional radiographic closure of the canal can be considered as a key factor in the long-term duration of the cure, as underlined by:

We added to the introduction the information about good quality obturation and its importance for controlling endodontic remission of disease as following:

"Biochemical preparation of the canal, the use of irrigation solutions, and root canal void free, three dimensional, hermetic filling that minimizes the opportunity of re-infection (12)are three factors used in the repair of apical periodontitis (13). "

-  Please check the references style, please following MDPI reporting instructions.

We checked the reference section according to MDPI instructions

Reviewer 2 Report

Dear Authors, 

please find the suggestions attached.

Author Response

In the materials and methods section:

- “The selected studies were assessed for risk of bias with parameters adapted from Sarkis-Onofre [18], Montagner [19] and AlShwaimi [20], to suit this review.” Why this choice? Please explain.  

We added the following to the discussion:

Different protocols have been purposed for risk of bias evaluation in in-vitro studies. We used a combination of three methods to standardized the bias assessment in order to provide the most comprehensive way to assess the risk of bias for this article

-We checked and changed the manuscript to reduce Double spaces

- In the discussion, I suggest that the authors underline an important context, represented by mechanical shaping, which still today represents one of the main ways of root canal disinfection, and which allows the use of simple 3D obturation techniques that are easy to perform and extremely rapid, as indicated by:

We added the following to the discussion:

Today along-side chemical irrigation, mechanical shaping is still one of the main ways of root canal disinfection which allows the use of easy and quick three dimensional obturation technique (37).

Reviewer 3 Report

Dear Authors, thanks for the work and the submission on this Journal. The review has been conducted properly, however there are several issues which need to be clarified.

1- Prospero guidelines are mentioned but there is no evidence in the reference. Maybe a table with the selection criteria would clarify better. Please do not report all the keywords in the text

2- At the end of the discussion you assess that no evidence is reported about using sonic and ultrasonic devices. Is this a bias? How can it affect your results?

3- what is the clinical relevance of this review?

Author Response

1- Prospero guidelines are mentioned but there is no evidence in the reference. Maybe a table with the selection criteria would clarify better. Please do not report all the keywords in the text

We added the citation of PROSPERO guidelines.

2- At the end of the discussion you assess that no evidence is reported about using sonic and ultrasonic devices. Is this a bias? How can it affect your results?

We added to the discussion the following:

"It is important to note that we only assessed studies that used needle irrigation, with no sonic or ultrasonic agitation. While it could introduce certain degree of bias, similar protocols of irrigation should be compared in order to prevent the heterogeneity of the evaluated studies in the systematic review (48)"

. 3- what is the clinical relevance of this review?

We added the following to the introduction:

". In daily clinical practice, the elimination of the bacteria within the root canal system is the most challenging aspect of the root canal treatment. Endodontic irrigants play an important role in the elimination of the bacteria. Various irrigation protocols have been purposed for this purpose without a necessary standardization. This review identifies the need to develop standardized methods to evaluate bacterial penetration depth after the use of different irrigants.. "

Round 2

Reviewer 1 Report

Dear Authors, 

Congratulations! I think you made a great work and this manuscript is now suitable for publication for me. 

Reviewer 3 Report

Dear Authors , thanks for correcting the manuscript. However I still believe that this study doesn’t improve the clinical practice , although it is well conducted on the formal point of view